# Nutritional Composition and Antioxidant Activity of *Gonostegia hirta*: An Underexploited, Potentially Edible, Wild Plant

**DOI:** 10.3390/plants12040875

**Published:** 2023-02-15

**Authors:** Yaochen Li, Zheng Hu, Xiaoqi Chen, Biao Zhu, Tingfu Liu, Jing Yang

**Affiliations:** 1Key Laboratory of Quality and Safety Control for Subtropical Fruit and Vegetable, Ministry of Agriculture and Rural Affairs, Collaborative Innovation Center for Efficient and Green Production of Agriculture in Mountainous Areas of Zhejiang Province, College of Horticulture Science, Zhejiang A&F University, Hangzhou 311300, China; 2School of Information and Electrical Engineering, Hangzhou City University, Hangzhou 310015, China; 3Lishui Academy of Agricultural Sciences, Lishui 323000, China

**Keywords:** *Gonostegia hirta*, nutritional value, chemical compounds, antioxidant activity

## Abstract

Wild, edible plants have received increasing attention as an important complement to cultivate vegetables, as they represent an easily accessible source of nutrients, mineral elements, and antioxidants. In this study, the tender stems and leaves of *Gonostegia hirta*, an edible species for which only scarce data are available in the literature, are thoroughly evaluated for their nutritional profile, chemical characterization, and antioxidant activity. Being considered as an underexploited, potentially edible plant, the nutritional composition of *Gonostegia hirta* was identified, and several beneficial compounds were highlighted: sugars, potassium, calcium, organic acids, fatty acids, phenolics, and flavonoids. A total of 418 compounds were identified by metabolomic analysis, including phenolic acids, flavonoids, amino acids, lipids, organic acids, terpenoids, alkaloids, nucleotides, tannins, lignans, and coumarin. The plant sample was found to have good antioxidant capacities, presented by DPPH, FRAP, ABTS^+^, hydroxyl radical scavenging capacity, and its resistance to the superoxide anion radical test. In general, *Gonostegia hirta* has a good nutritional and phytochemical composition. The health benefits of *Gonostegia hirta* as a vegetable and herbal medicine is important for both a modern diet and use in medication.

## 1. Introduction

Wild, edible plants are an important plant resource, not only as an inexpensive source of nutrients, vitamins, antioxidants, and functional ingredients because of their nutritional potential, but also as a cultural heritage practice that should be preserved. In some less-developed regions and economically disadvantaged areas in the world, wild, edible plants are still traditionally consumed and play an important role in the diet of local populations.

Globally, about 30,000 edible plant species are recorded at present; however, only about 0.5% of them are widely grown [1]. The disappearance of many wild, edible plant species has been exacerbated by an over-reliance on high-yielding and genetically consistent crops. This has led to a monoculture of edible plants that can be grown on a large scale and can be put into production [2]. To date, global agriculture products do not seem to be sufficient enough to meet human’s nutritional needs. Meanwhile, wild, edible plant species that can be collected from the wild or grown using traditional methods can potentially play a role in diversifying diets and combating “hidden hunger” caused by micronutrient deficiencies [3]. In recent years, wild, edible plants have not only received further attention from the scientific community, with many wild, edible plants being discovered, documented, and studied for their chemical and functional properties, but also from the food industry and consumers, who are increasingly interested in sustainable and healthy foods. In addition, in some countries and regions, such as Europe, Japan, and China, local-specialty wild plants are transformed into various products, dishes, and skin care products as cultural promotion and exchange practices, while also seeking their health benefits.

*Gonostegia hirta* is a perennial herb of the family Urticaceae, and is a common Chinese medicinal product and wild, edible vegetable. On the one hand, *Gonostegia hirta* is an ethnic medicine commonly used in the treatment of stomachaches, bleeding, and mastitis [4]. A few studies have shown that *Gonostegia hirta* contains flavonoids, phenols, organic acids, polysaccharides, and other bioactive substances, while flavonoids, phenols, and organic acids have strong antioxidant properties [5,6]. An ethnobotanical survey of wild plants was conducted by Hong et al. [7] and indicted that *Gonostegia hirta* is useful as a traditional medicinal herb, and it also can be consumed as herbal tea after being dried, which has the effect of clearing heat and detoxifying the body, spleen tonic, digestion, and removing dampness.

On the other hand, *Gonostegia hirta* is also often used as an edible, wild plant. Of the different parts of *Gonostegia hirta* that may have a nutritional value (the leaves, young stems, and roots), the leaves and young stems are the most frequently consumed parts. These leaves and young stems can be eaten in stir-fries, cooked in soups, or made into dumplings. Although it is worthy of being noticed as both a nutritional source and traditional medicine plant, the relevant research focusing on the chemical characterization of *Gonostegia hirta* is still inadequate, and few reports have been conducted on its nutritional composition. The present study has four main objectives concerning the nutritional value and chemical characteristics of *Gonostegia hirta*: (1) to characterize the main nutritional values obtained from moisture, crude fat, soluble sugars, fatty acids, amino acids, minerals, and organic acids; (2) to evaluate the safety concerning the total content of potentially toxic and toxic elements and total saponins; (3) to evaluate antioxidant and potential functional properties through chemical content and metabolomics analysis; and (4) to comprehensively evaluate the nutritional, flavor, and functional values to meet the needs of healthy, vegetable diets to provide a basis for the development and utilization of *Gonostegia hirta*.

## 2. Results and Discussion

### 2.1. Chemical Composition

#### 2.1.1. Moisture and Crude Fat

Table 1 shows that the moisture content of *Gonostegia hirta*, 50.84%, and its crude fat content, 2.57%. The fat content of vegetables is usually very low; therefore, the low-fat content observed in *Gonostegia hirta* supports this fact.

#### 2.1.2. Minerals and Potentially Toxic Elements

As shown in Table 1, the highest amounts of minerals observed in *Gonostegia hirta* are potassium (K), calcium (Ca), phosphorus (P), and magnesium (Mg). Their contents were 3855.22, 1352.49, 569.51, and 284.35 mg/100 g of dry weight (DW), respectively. K is an essential nutrient that contributes to maintain total body fluid, acid and electrolyte balance, and normal cell function. Increasing K intake may be beneficial for most people in preventing and controlling elevated blood pressure levels and strokes [8]. Ca is also important for controlling blood pressure and promoting bone and tooth growth [9]. P plays a vital role in energy generation, human growth and development, and also provides the structural framework for DNA and RNA [10]. It also functions in regulating blood sugar levels [11]. Mg has numerous functions, including cell signaling and energy production, and it is also an important mineral in bones, cell membranes, and chromosomes [9]. The content of Ca and Mg in *Gonostegia hirta* is much higher than in some cultivated vegetables. For example, the Ca content in *Gonostegia hirta* was 5.12- and 22.24-times higher than in cassava leaves and chicory [12,13], respectively, and the Mg content in *Gonostegia hirta* was 7.10-, 5.10-, and 9.38-times higher than in chicory, green lettuce, and Swiss chard, respectively [13]. It has been suggested that *Gonostegia hirta* could act as an important supplement for calcium and magnesium. Zinc (Zn) has many roles in the body, including immune function, growth and development, nerve function, vision, and fertility [9]. Compared to spinach (7.95 mg/100 g DW) [14] and amaranth (3.7 mg/100 g DW) [12], *Gonostegia hirta* also has a higher Zn content (10.47 mg/100 g DW). However, it is important to note that the excessive consumption of zinc (estimated dose of 325–650 mg) is dangerous to human health [15]. Minerals, such as K, Ca, Mg, P, and Zn, are crucial for normal body development and maintenance. The results indicate that *Gonostegia hirta* is rich in these essential minerals, of which Fe, Zn, Ca, Mg, and Cu are some of the most commonly deficient mineral elements in the human diet [11]; thus, *Gonostegia hirta* seems to be a multifaceted supplement for these elements in the human body.

The contents of cadmium (Cd), lead (Pb), and chromium (Cr) were also measured to evaluate the safety of *Gonostegia hirta*, and the results show the Cd, Pb, and Cr contents in *Gonostegia hirta* are within the safe range, according to the Commission Regulation (EU) 2021/1317, Commission Regulation (EC) No 1881/2006, and Chinese National Food Safety Standard for Maximum Levels of Contaminants in Foods (GB 2762-2017). In general, *Gonostegia hirta* is less contaminated by potentially toxic and toxic elements; however, it might also be associated with the local climate, soil, and ecological environment, and should not be picked in contaminated areas, such as areas with developed industries, high traffic pollution, and high sewage discharge [16].

#### 2.1.3. Soluble Sugars and Organic Acids

In general, monosaccharides are easily digested and absorbed by the human body; therefore, they are thought to have a high nutritional value [17]. Three types of soluble sugars were measured in *Gonostegia hirta*, namely, sucrose, glucose, and fructose. Among these, fructose has the highest value, accounting for 46.71% of the total (Table 1), and is sweeter and tastier than sucrose and glucose [18]. The total soluble sugar content of *Gonostegia hirta* is 87.03 mg/g, which higher than lettuce (7.72 to 16.24 mg/g DW) [19] and very close to soybean (84.70 to 140.91 mg/g) [20].

Organic acids affect a number of characteristics of vegetables, such as flavor, aroma, and appearance [21]. The most abundant organic acid detected in *Gonostegia hirta* was citric acid (22.5 mg/g DW), followed by tartaric (10.68 mg/g DW), oxalic (3.4 mg/g DW), and ascorbic (1.30 mg/g DW) acids (Table 1). Among these, citric acid is an important organic acid with a strong, sour taste, is easily soluble in water, and is a natural preservative [22], while tartaric acid is a natural antioxidant and flavorant in foods [23]. Moreover, oxalic acid easily combines with mineral elements to form insoluble oxalate, and vegetables with a high oxalic acid content can be consumed in large quantities with a distinct bitter taste and may cause hyperoxaluria and kidney stones [24,25]. *Gonostegia hirta* contains slightly higher oxalic acid levels than lettuce (1.16 to 1.67 mg/g DW) [18], but less than conventional spinach (about 8 to 12.5 mg/g) [26]. Thus, it is recommended to blanch *Gonostegia hirta* to reduce the bitterness caused by oxalic acid when consuming it. In addition, *Gonostegia hirta* has a higher ascorbic acid content compared to spinach and chicory, and the ascorbic acid levels are 0.52 mg/g of fresh weight (FW) and 0.30 mg/g FW [13]. Ascorbic acid (vitamin C) is considered one of the most powerful and least toxic, natural antioxidants, which has been shown to be effective against superoxide radical anions, H_2_O_2_, hydroxyl radicals, and singlet oxygen [27].

In total, 27 organic acid species were detected by Ultra Performance Liquid Chromatography (UPLC)—tandem mass spectrometry (MS/MS), accounting for 6.46% of 418 metabolites in total (Figure 1 and Appendix A). Some of these organic acids are thought to have unique functions, such as succinic acid (SA), mangiferic acid, and kynurenic acid (KYNA). SA is an important metabolite that has been shown to be involved in promoting energy expenditure and fighting obesity [28]. Mangiferic acid can promote human skin regeneration, and it is expected to be used as a new material in skin regeneration biotherapy or cosmetics in the future [29]. KYNA is a metabolite of kynurenine, which has anti-inflammatory, antioxidant, and pain-relieving properties [30].

#### 2.1.4. Amino Acid

Amino acid (AA) composition is a reliable indicator of the nutritional value of food and is traditionally classified as essential amino acids (EAAs) and non-essential amino acids (NEAAs). As shown in Table 2 and Figure 2, 17 amino acids were detected in *Gonostegia hirta*, and 8 of them present EAAs, namely, isoleucine (Ile), leucine (Leu), methionine (Met), lysine (Lys), valine (Val), threonine (Thr), phenylalanine (Phe), and histidine (His), which have an important effect on the nutritional quality of plant-based products. His has the highest content (0.091 mg/g DW), followed by Thr (0.033 mg/g) and Val (0.020 mg/g DW). The content of Met in *Gonostegia hirta* is 0.015 mg/g DW, and it is a sulfur-containing amino acid that is essential for cellular metabolism and protein uptake. In addition, aspartic acid (Asp) is the most abundant of all 17 amino acids, which may contribute to the fresh taste of *Gonostegia hirta*.

The content and ratio of individual amino acids are also related to the flavor of the plant. For example, Asp and glutamic acid (Glu) are thought to be significant contributors to the fresh flavor [30], which accounts for 32.06% of total amino acids in *Gonostegia hirta*. At the same time, the proportion of sweet amino acids (Thr + Ser + Gly + Ala + Lys + Pro) [31] was 26.22%; the proportion of bitter amino acids (Val + Met + Ile + Leu + Phe + His + Arg) [31] was 20.63%. The results show that the content of bitter amino acids is lower than sweet and fresh amino acids, indicating that the palatability of *Gonostegia hirta* might be relatively good. In addition, some amino acids are considered as medicinal amino acids, including Asp, Glu, Gly, Met, Leu, Tyr, Phe, Lys, and Arg [31], which account for 62.19% of the total amino acids in *Gonostegia hirta.* For example, Tyr may improve cognitive ability for human adults, while Arg has a beneficial effect on the regulation of nutrient metabolism to enhance lean tissue deposition and insulin resistance in humans [32,33]. Additionally, those two compounds had the highest levels of all medicinal amino acids.

Some amino acid derivatives were also detected by UPLC-MS/MS, such as *trans*-4-hydroxy-l-proline (Hyp), *N*-Acetylneuraminic acid (NANA), L-theanine, proline betaine, and pipecolic acid (Appendix A). These amino acid derivatives have different roles in the growth and metabolism of plants, medicinal functions, etc. For example, Hyp is a useful chiral building block for the production of many nutritional supplements and drugs [34]. NANA consumes toxic hydrogen peroxide under physiological conditions and may serve as an intrinsic antioxidant in the future [35]. L-theanine is known to be an amino acid unique to tea, and has positive effects on relaxation, cognitive performance, emotional state, sleep quality, cancer, cardiovascular disease, obesity, and the common cold [36].

#### 2.1.5. Fatty Acid

The composition of fatty acids has a significant impact on mammalian health. For example, some saturated fatty acids (SFAs) have been reported to exhibit deleterious effects, while the intake of monounsaturated fatty acids (MUFAs) and polyunsaturated fatty acids (PUFAs) reduced the risk of cardiovascular and other metabolic diseases, thereby improving one’s lifespan [37]. Therefore, lipid composition is critical for evaluating the quality of wild, edible plants. The fatty acid profile of *Gonostegia hirta* is shown in Table 3, evidencing that a total of 16 fatty acids can be identified (Figure 3). The two major compounds are the essential fatty acids linolenic and linoleic acids, with SFAs accounting for about 27.4% and PUFAs accounting for about 72.6%. In the samples, PUFAs were the major group, with ω-3 fatty acids accounting for 31% of total fatty acids. Previous studies suggested that a higher intake of ω-3 may protect us from inflammatory diseases, cancer, cardiovascular disease, and other chronic diseases [38]. It has been reported that the optimal intake ratio of ω-6 to ω-3 fatty acids is 1~2/1 [39], and the ratio of ω-6 and ω-3 fatty acids in *Gonostegia hirta* is 1.32/1, which means the consumption of *Gonostegia hirta* could possibly be beneficial for our health.

Except for the 16 fatty acids mentioned above, more fatty acids were detected by UPLC-MS/MS, and some of them, such as elaidic and punic acids, serve many important roles. Ohmori et al. observed that elaidic acid may provide significant metastatic potential for colorectal cancer (CRC) cells, which has important implications for the treatment of CRC [40]. Punicic acid has great potential as an antioxidant, anti-diabetic, and natural healing agent for inflammatory diseases [41]. In addition to these fatty acids, we also found a phosphatidylcholine called choline alfoscerate; it is a common choline compound and acetylcholine precursor in the brain that has been shown to be effective in the treatment of Alzheimer’s disease and dementia [42].

#### 2.1.6. TP (Total Phenols), TF (Total Flavonoids), and TS (Total Saponins)

Phenols are widely distributed in nature and their anti-microbial, anti-bacterial, antioxidant, pharmacological, and nutritional properties have been widely recognized [43]. The TP content of *Gonostegia hirta* was determined (Table 4). The result indicates that *Gonostegia hirta* is a good source of phenolics (TP: 1.20 ± 0.10 mg GAE/g DW), with an amount 2.43-times higher than spinach (0.493 mg GAE/g DW) and 5.53-times higher than green lettuce (0.217 mg GAE/g) [13]. However, the result in this study is lower than that of Wong et al. (5.6 and 5.3 mg GAE/g DW for leaves and stems, respectively) [5]. The reason for the discrepancy may be due to the plant development stage, picking regions, and extraction methods. In addition, 47 phenolic acid compounds were found by UPLC-MS/MS, accounting for 11.24% of the total (Figure 1). *p*-Coumaric, ferulic, caffeic, chlorogenic, and coumaric acids were included, which received a lot of attention in the previous study, and are considered to have a variety of active functions [44]. For example, extensive studies have shown that *p*-Coumaric acid (especially the *p*-CA adduct) presents a variety of biological activities, including antioxidant, anti-inflammatory, anti-platelet, and anti-cancer activities, as well as reducing atherosclerosis, various tissue damage, neuronal damage, gout, and diabetes [45]. Ferulic acid has a good antioxidant capacity due to the presence of phenolic hydroxyl groups with a hydrogen-donating capacity [46]. The synergistic effect of these substances provides a certain basis for the potential biological activity of *Gonostegia hirta*.

The TF content in *Gonostegia hirta* was also measured in the present study, and a value of 76.49 ± 5.58 mg/g was obtained, which is 2.85-times higher than that of Brassica juncea [47]. To date, more than 6000 flavonoids have been identified, and their basic functions in plants include regulating growth and providing protection against pathogens. Because of their antioxidant, antitumor, anti-inflammatory, antibacterial, and antiviral activities, they have attracted considerable interest [48]. Flavonoids were the most diverse of the 418 identified metabolites, accounting for 31.58% (Figure 1). The flavonoids identified can be divided into ten main groups, as shown in Figure 1. Quercetin, kaempferol, popcornin, apigenin, and lignan are the five most prevalent plant flavonoids, which were all detected in *Gonostegia hirta.* Apigenin has low toxicity and multiple beneficial bioactivities, such as anti-cancer and antibacterial effects [49,50], and received a considerable attention. Lignans, quercetin, prunetin, and kaempferol have also shown strong anti-cancer effects in various tests [51,52]. In this study, kaempferol, quercetin, isorhamnetin-3-*O*-rutinoside, and kaempferol 3-*O*-rutinoside were detected, and this is consistent with the results of Lei et al. [53].

Saponins are sugar-conjugated natural compounds with a variety of biological properties, such as anti-inflammatory, anticancer, antioxidant, and immunomodulatory [54]. Nevertheless, it has been shown that high doses of saponins can cause damage to the liver [55]. A TS content of 60.30 ± 0.66 mg/g was detected in *Gonostegia hirta,* and suggested that the value was within the range compared to *Aralia elata* (38.37 to 104.31 mg/g DW) [56]. The metabolites of *Gonostegia hirta* were detected by UPLC-MS/MS and three terpenoids were identified, including soyasapogenol B, 3, 24-Dihydroxy-17, 21-semiacetal-12(13), oleanolic fruit, and 2-Hydroxyoleanolic acid (Appendix A). Of the abovementioned compounds, only soyasapogenol B belongs to triterpene saponins, which have also been shown to have a good ability to inhibit the proliferation of human liver cancer cells [57].

#### 2.1.7. Alkaloids

Alkaloids are the main chemical constituents and the basis of medicinal substances in many medicinal plants, and some alkaloid components are also the main substance basis for the toxicity of plants [58]. The present study detected 23 alkaloids (Appendix A), among which trigonelline, betaine, and caffeine are thought to be the most important substances. Trigonelline has a bitter taste and has been reported to have hypolipidemic, antiviral, central nervous system therapeutic, and memory-retention effects [59,60]. Although some studies suggest it has low toxicity levels, Mishkinsky et al. reported that oral and subcutaneous LD_50_ of trigonelline in rats of was ≈ 5000 mg/kg, and no mice died during the experiment, even when 50 mg/kg of fenugreek was fed to them for 21 days. Therefore, it is safe to consume vegetables that contain trigonelline, such as *Gonostegia hirta*, in moderation [61]. Some studies have shown that betaine has anti-inflammatory and antioxidant properties, and is also beneficial to alleviates endoplasmic reticulum stress, and therefore contributes to improve insulin sensitivity and better glucose clearance, with a greater potential to fight diabetes [62]. Caffeine is naturally found in various foods, such as coffee, tea, and cocoa, and has also been detected in *Gonostegia hirta*. Caffeine is used in cosmetics for its high antioxidant capacity and ability to penetrate the skin barrier [63], and it is often used as a stimulant [64]. An allergen was also detected, namely, cocamidopropyl betaine, which is mainly used as a surfactant in cosmetics, and it may cause a skin allergy on the head and neck. Thus, the subsequent use of *Gonostegia hirta* requires attention [65,66]. The types of alkaloids present in *Gonostegia hirta* can also be divided into phenolamine and indole alkaloids, the more useful of which is spermine, which has been shown to be associated with the development of various human cancers and can be used as a diagnostic, prognostic, and therapeutic tool for various cancers [67].

### 2.2. Antioxidation Capacity Analysis

To date, only a few studies have been conducted on the antioxidant properties of *Gonostegia hirta*. The values of the DPPH and ABTS^+^ scavenging capacities of *Gonostegia hirta* were 5.92 and 68.82 mg/g, respectively (Table 5). Additionally, the values are much higher than for cabbage (1.04 mg/g DW and 0.28 mg/g FW) [68,69]. However, the ferric-reducing power (62.23 μmol/L) of *Gonostegia hirta* is less than for cabbage (53.01 μmol/g) [68]. This may be due to the differences in antioxidant capacity-determination methods. The scavenging ability of DPPH and ABTS radicals is based on a combination of hydrogen atom transfer (HAT) and single electron transfer (SET) mechanisms, whereas FRAP is based on a SET mechanism, and the whole reaction does not involve free radicals, but only electron transfer ability. This may lead to the difference in the ability reflected by FRAP, compared to the other two components [70].

In addition, ABTS radicals are soluble in water, and organic solvents and are mainly used to determine the antioxidant capacity of lipophilic and hydrophilic compounds (e.g., vitamin C, vitamin E, phenolic compounds, and anthocyanins). However, DPPH radicals are insoluble in water and are usually soluble in methanol, ethanol, or their aqueous mixtures (water content should not exceed 60%), and the antioxidant capacity of scavenging DPPH radicals is correlated with phenolic acids and flavonoids [70].

Therefore, we speculated that the DPPH and ABTS^+^ scavenging abilities of *Gonostegia hirta* may be related to the abundant phenolic substances, such as caffeic acid, ferulic acid, p-coumarins, and quercetin, which are thought to have high free-radical scavenging capacities, all of which were detected in *Gonostegia hirta* [71,72,73]. Compared to some common vegetables [13,47], *Gonostegia hirta* has higher TP and TF levels. In general, the higher the content of TP and TF, the greater the antioxidant activity shown by the plant [74], which may also indicate its good antioxidant capacity. In addition, the presence of ascorbic acid also significantly contributes to the antioxidant capacity of plants. We can also observe, compared to other studies, the ascorbic acid content of *Gonostegia hirta* is also much higher than that of some other vegetables, such as spanich, chicory, and green lettuce [13,26]. It is suggested that the antioxidant capacity of *Gonostegia hirta* might be better and could contain more substances with an antioxidant activity. The chemical composition of the plant significantly changes with its growth and development, which can be some of the reasons for the difference in its chemical composition, compared to other vegetables.

## 3. Materials and Methods

### 3.1. Plant Material

*Gonostegia hirta* was obtained from Li Shui, Zhejiang province (27°25′ N~28°57′ N, 118°41′ E~120°26′ E) on 19 July 2019. Fresh stems and leaves were washed with ionized water, freeze-dried, crushed, and then stored at −20 °C until further analysis.

### 3.2. Chemical Composition Measurement

#### 3.2.1. Moisture

The samples were cleaned and weighed (W1), then freeze-dried for 72 h and weighed again (W2), and the moisture percentage was calculated as:$$\mathrm{Moisure}\;\left(\%\right)=\frac{W1-W2}{W1}\times 100$$

#### 3.2.2. Crude Fat

Crude fat was determined as previously described by Xu. [75]. Fatty acids in the sample were extracted with 10 mL of hexane (1.0 g sample) (W1). The leaves were sonicated in a water bath at 42 °C for 10 min, centrifuged, and filtered. A total of 3 extractions were performed. The combined extracts were rotary evaporated and oven dried at 60 °C to a constant weight (W2), and the crude fat percentage was calculated as:$$\mathrm{Crude}\;\mathrm{fat}\;\left(\%\right)=\frac{W2}{W1}\times 100$$

#### 3.2.3. Minerals and Potentially Toxic Elements

The elements were determined as previously described by Yang et al. [76]. Briefly, the freeze-dried powder (0.50 g) was digested with concentrated HNO_3_, fixed with water to 50 mL, and analyzed for minerals with a Inductively Coupled Plasma Optical Emission Spectrometer (ICP-OES). We used an IRIS/AP-ICP (TJA, Dartmouth, MA, USA) instrument and the ICP-OES procedure mainly referenced the “Food National Standard for Determination of Multi-element in Food” (GB 5009. 268-2016).

#### 3.2.4. Soluble Sugar

Soluble sugars were determined by high-performance liquid chromatography coupled to a refraction index detector (HPLC–RID), mainly referring to the method of Yao [77]. The samples were extracted (0.1000 g) with 6 mL of ultrapure water in a water bath at 65 °C for 20 min. The extraction solution was filtered by a 0.45 μm microporous filter prior for analysis.

A sample of 5 μL was analyzed in an Agilent 1200 HPLC system equipped with a differential refractive index detector, using a Waters sugar-Pak I column WAT084038 (6.5 × 300 mm, 10 µm). The column temperature was maintained at 85 °C at a rate of 0.8 mL/min, and the mobile phase was ultrapure water. The contents of glucose, sucrose, and fructose in the samples were calculated according to the standard curves drawn from the results of the injection.

#### 3.2.5. Organic Acid

Reducing the organic acid was determined according to the method described by Song et al. [78]. The extract was obtained by passing 0.2000 g of sample and 10 mL of KH_2_PO_4_-H_3_PO_4_ buffer solution (0.02 mol/L, pH = 2.9) in a water bath at 75 °C for 1 h. The extraction solution was filtered by a 0.45 μm microporous filter prior for analysis. The results are expressed as mg per g of plant DW.

The samples were analyzed on an Agilent 1200 HPLC system equipped with a DAD detector using column InertSustain C18 (4.6 mm × 250 mm, 5 μm). The mobile phase was KH_2_PO_4_-H_3_PO_4_ at a rate of 0.8 mL/min. The contents of oxalic, tartaric, ascorbic, and citric acids in the samples were calculated from the standard curves drawn from the injection results.

#### 3.2.6. Amino Acid

The extracts were prepared using 0.1000 g of sample and 5 mL of ultrapure water. The samples were extracted in an ultrasonic bath at room temperature for 60 min. The extract was filtered through a 0.22 μm filter, and then stored at 4 °C prior to analysis.

A total of 10 μL of sample solution, 70 μL of AccQ buffer, and 20 μL of derivatives were mixed for 15 s and heated in an oven at 55 °C for 10 min. The samples were then analyzed using the AccQ Tag system (Waters, Milford, MA, USA) by Waters Arc HPLC with an AccQ TagTM amino acid column (100 mm × 2.1 mm, 1.7 μm, Waters Corporation, Milford, MA, USA). The rate was 1 mL/min. Mobile phase A was AccQ-Tag A solution diluted at 1:10 (*v/v*) with ultrapure water, mobile phase B was acetonitrile, and mobile phase C was ultrapure water. The content of each amino acid quantity in the sample was calculated from the standard curve drawn from the injection results.

#### 3.2.7. Fatty Acid

Fatty acids were determined as previously described by Xu. [75], and GC-MS methods mainly referenced the “Food National Standard for Safety Determination of Fatty Acids in Foods” (GB 5009.168-2016). The extracts (Section 3.2.2) were 2 mL of hexane and 20 μL of undecanoic acid standard (1 mg/mL), mixed and analyzed by GC-MS (GCMS-QP2010 SE, SHIMADZU) after methylation with the KOH-CH_3_OH and H_2_SO_4_—CH_3_OH solutions.

#### 3.2.8. Total Phenolic Content (TPC), Flavonoid Content (TFC), and Saponins (TS)

The total phenol content was determined using the Folin–Ciocalteau method, according to Colonna et al. [13], with some modifications. The extracts were prepared using 0.1000 g of dry, raw material and 10 mL of 80% methanol. In a test tube, 0.6 mL of extract, 3 mL of Folin–Ciocalteu reagent, and 2.0 mL of Na_2_CO_3_ 7.5% were added. After the mixture was vortexed, it was allowed to react for 60 min in the dark. Finally, the absorbance at 765 nm was measured with a spectrophotometer (UV-2600, SHIMADZU, Tokyo, Japan) using distilled water as a blank. The content of total phenols was expressed as equivalent mg of gallic acid per g of the dry sample.

Total flavonoid content was estimated by the AlCl_3_ colorimetric method [47]. Briefly, 20 mg/mL of the extract was prepared using 60% ethanol. The extract of 2 mL was used with 0.4 mL of a 10% (*p*/*v*) aluminum trichloride (AlCl_3_) solution and 0.4 mL of 5% sodium nitrate (NaNO_2_). Subsequently, the mixture was vortexed and incubated for 6 min. Then, 4 mL of NaOH (4%) was added to stop the reaction. Finally, the absorbance at 510 nm was measured with a spectrophotometer (UV-2600, SHIMADZU) using 60% ethanol as a blank. The total flavonoid content was expressed as equivalent mg of rutin per g of the dry sample.

Total saponins (TSs) were quantified by the colorimetric method according to Le et al. [17], with some modifications. The extracts were prepared using 0.1000 g of dried plant tissue and 7 mL of 70% (*v/v*) ethanol, and sonicated at 55 °C for 40 min. In a test tube, evaporate 0.2 mL of the extract in a water bath at a temperature of 70 °C and add 0.1 mL of 5% vanillin reagent and 0.4 mL of perchloric acid. The reaction was heated in a water bath at 60 °C for 15 min, and ethyl acetate (4 mL) was added for 10 min. Finally, the absorbance at 560 nm was measured with a spectrophotometer (UV-2600, SHIMADZU, Tokyo, Japan) using 70% ethanol as a blank. The total saponin content was expressed as equivalent mg of oleanolic acid per g of the dry sample.

### 3.3. Metabolite Composition Identification

The metabolite components were identified using the UPLC-ESI-MS/MS method, mainly based on Chen et al.’s study [79], with some modifications. A total of 100 mg of powder ground from each lyophilized leaf was weighed and extracted overnight at 4 °C with 0.6 mL of extracting solution. After centrifugation at 10,000× *g* for 10 min, the supernatant was aspirated. Samples were analyzed using a Waters ACQUITY UPLC HSS T3 C18 (1.8 μm, 2.1 mm × 100 mm) by a UPLC-ESI-MS/MS system (UPLC, Shim-pack UFLC CBM30A system, SHIMADZU; MS, 4500 Q TRAP, Applied Biosystems, Waltham, MA, USA) to analyze 4 μL samples. The mobile-phase solvent A was ultrapure water (containing 0.04% acetic acid) and solvent B was acetonitrile (containing 0.04% acetic acid). The column temperature was 40 °C and a gradient procedure was used for sample measurements. The effluent was alternately connected to an ESI-triple quadrupole-linear ion trap (QTRAP)-MS. The ESI source operating parameters were set as follows: electrospray ion source temperature, 550 °C; mass spectrometry voltage, 5500 V; curtain gas set to 30.0 psi; and collision-activated dissociation (CAD) high. Raw files generated from UPLC-ESI-MS/MS analysis were analyzed with Analyst 1.6.3 software. Q3 was used for metabolite quantification, while Q1, Q3, RT (retention time), DP (declustering potential), and CE (collision energy) were used for metabolite identification.

### 3.4. Antioxidant Activity Evaluation

The antioxidant activity of methanolic extracts was measured in vitro and included the following assays: DPPH radical scavenging activity, ABTS^+^ scavenging capacity, iron ion-reducing capacity, hydroxyl radical scavenging capacity, and anti-superoxide anion capacity. DPPH radical scavenging activity, ABTS^+^ scavenging capacity, and ferric ion-reduction capacity refer to Jia et al. [80] with slight modifications. Hydroxyl radical scavenging capacity and resistance to the superoxide anion radical (O_2_^−^) test were conducted with a hydroxyl free-radical assay kit and inhibition and production superoxide anion assay kit (Nanjing Jiancheng Bioengineering Research Institute).

DPPH radical scavenging activity: 2.7 mL of DPPH methanol solution (0.1 mmol/L) was added to the methanol extract (0.3 mL), as presented in Section 3.2.8, and the reaction was monitored at 517 nm until the absorbance was constant. Measurements were repeated three times for each sample. DPPH radical scavenging capacity was expressed as the equivalent amount of ascorbic acid (vitamin C) per gram of sample.

ABTS^+^ scavenging capacity combined 3.8 mL of ABTS^+^ working solution and extract (0.2 mL) and monitored the reaction at 734 nm. The ABTS^+^ radical scavenging capacity was expressed as the amount of ascorbic acid (vitamin C) equivalent per gram of sample.

Iron ion-reducing capacity: 1.8 mL of TPTZ working solution (100 mL of 0.3 mol/L acetate buffer; 10 mL of 10 mmol/L TPTZ solution and 10 mL of 20 mmol/L FeCl_3_ solution, a mixture of the three solutions) was added to 3.1 mL of ultrapure water and 0.1 mL of the extract to be tested, and the absorbance of the mixture was measured at 593 nm. All analyses were performed in triplicate. The millimolarity of FeSO_4_ was expressed as its reducing power.

### 3.5. Statistical Analysis

The values of all indexes determined in the experiment were recorded in triplicate and were expressed as the mean ± standard deviation. SPSS Software (version 26.0) was used to apply these statistical tools.

## 4. Conclusions

The nutritional composition of *Gonostegia hirta* was evaluated and we concluded that it can be an important nutrient supplement, K, Ca, soluble sugars, organic acids, Asp, and omega-3 fatty acids. Moreover, the low content of potentially toxic and toxic metals (Pb, Cd, and Cr) indicated that the area from which the plant was removed was not contaminated. Additionally, *Gonostegia hirta* presented a high content of TP, TF, and TS, and contained a variety of health-promoting phytochemicals, such as rutin, caffeic acid, ferulic acid, vitamin C, vitamin E, etc, which proves that it can be an important source of natural antioxidants and has great potential to become a functional ingredient in the future. Furthermore, its methanol extract showed that it exhibited very good antioxidant abilities in DPPH, ABTS^+^, FRAP, hydroxyl radial scavenging capacity, and resistance to superoxide anion radical assays. In summary, a comprehensive analysis of *Gonostegia hirta* will help us to strengthen the protection and understanding of underexploited, edible, wild plants, and promote their growth, while simultaneously highlighting the interest in this edible, wild plant as a healthy dietary supplement.

## Figures and Tables

**Figure 1 plants-12-00875-f001:**
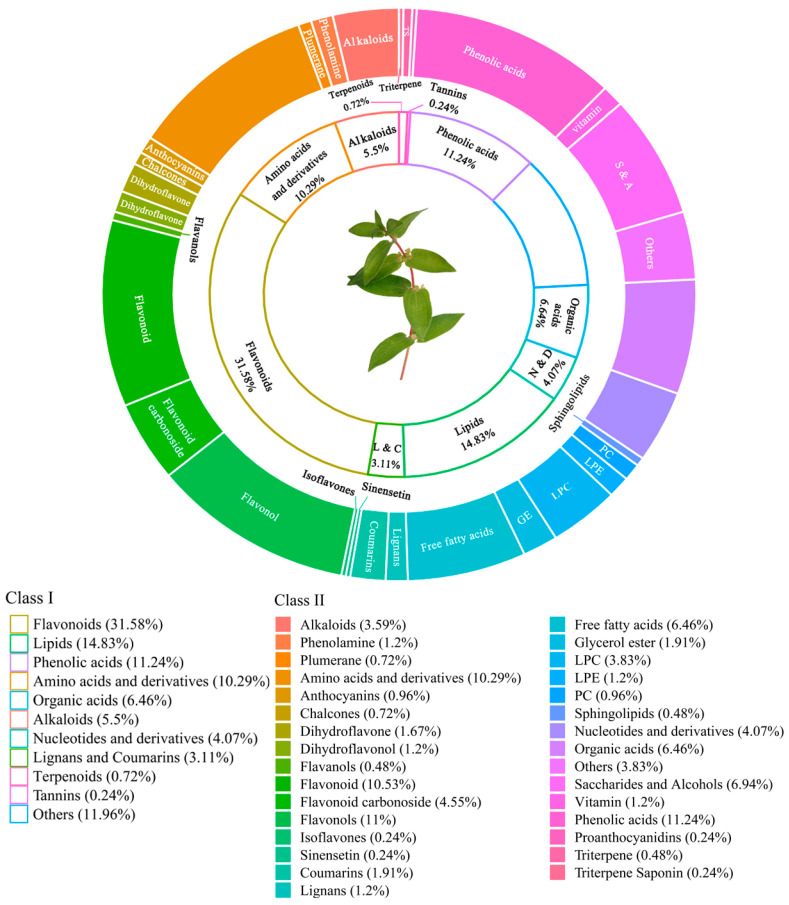
Percentage of metabolite substance types of *Gonostegia hirta* by *UPLC-MS/MS*. The inner circle shows the 11 major groups into which the 418 metabolites are divided, namely, flavonoids, lipids, phenolic acids, amino acids and derivatives, organic acids, alkaloids, nucleotides and derivatives (N and D), lignans and coumarins (L and C), terpenoids, tannins, and others. The outer circle is a subdivision of the 11 major groups, namely, alkaloids, phenolamine, and plumerane of alkaloids; anthocyanins, chalcones, dihydroflavone, dihydroflavonol, flavanols, flavonoid, flavonoid carbonoside, flavonols, and isoflavones of flavonoids; sinensetin, coumarins, and lignans of L and C; free fatty acids, glycerol ester (GE), lysophosphatidylcholine (LPC), lysophosphatidylethanolamine (LPE), phosphatidylcholine (PC), and sphingolipids of lipids; N and D and organic acids; saccharides and alcohols (S and A) and vitamin of others; phenolic acids and proanthocyanidins of tannins; triterpene and triterpene saponin (TS) of terpenoids.

**Figure 2 plants-12-00875-f002:**
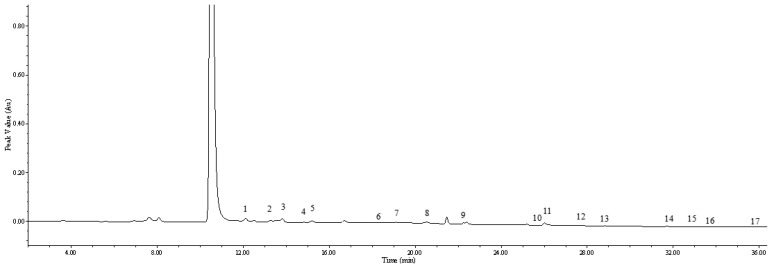
HPLC chromatogram of amino acids in *Gonostegia hirta*: 1. Asp; 2. Ser; 3. Glu; 4. Gly; 5. His; 6. Arg; 7. Thr; 8. Ala; 9. Pro; 10. Cys; 11. Tyr; 12. Val; 13. Met; 14. Lys; 15. Ile; 16. Leu; and 17. Phe.

**Figure 3 plants-12-00875-f003:**
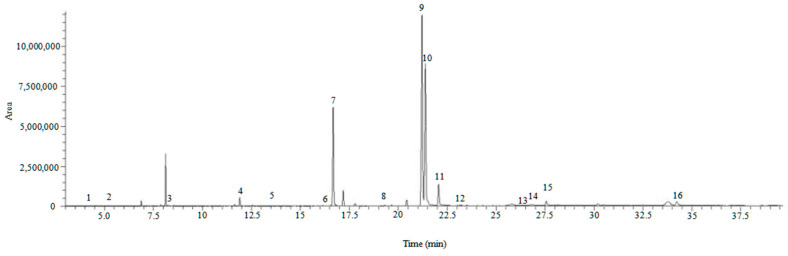
GC-MS chromatogram of fatty acids in *Gonostegia hirta*: 1. Octanoic acid; 2. Decanoic acid; 3. Lauric acid; 4. Myristic acid; 5. Pentadecanoic acid; 6. Palmitoleic acid; 7. Palmitic acid; 8. Heptadecanoic acid; 9. Linoleic acid; 10. Inolenic acid; 11. Stearic acid; 12. Stearolic acid; 13. Nonadecanoic acid; 14. Arachidonic acid; 15. Eicosanoic acid; and 16. behenic acid.

**Table 1 plants-12-00875-t001:** The contents of moisture, crude fat, mineral elements, soluble sugars, and organic acids in *Gonostegia hirta*.

	Compounds	Content
Moisture (%)	-	50.84 ± 0.01
Crude fat (%)	-	2.57 ± 0.03
Minerals (mg/100 g DW)	Calcium	1352.49 ± 52.19
	Potassium	3855.22 ± 88.35
	Magnesium	284.35 ± 11.51
	Phosphorus	569.51 ± 15.84
	Aluminium	10.37 ± 0.41
	Cuprum	0.75 ± 0.01
	Ferrum	9.96 ± 0.28
	Zinc	10.47 ± 0.37
Potentially toxic elements (mg/100 g DW)	Cadmium	0.02 ± 0.00
	Lead	0.06 ± 0.01
	Chromium	0.09 ± 0.03
Soluble sugars (mg/g DW)	Sucrose	26.89 ± 0.87
	Glucose	19.49 ± 1.34
	Fructose	40.65 ± 1.64
Organic acids (mg/g DW)	Oxalic acid	3.4 ± 0.78
	Tartaric acid	10.68 ± 1.72
	Ascorbic acid	1.30 ± 0.14
	Citric acid	22.50 ± 3.82

Note: “-”: does not have other compounds; “DW”: dry weight.

**Table 2 plants-12-00875-t002:** Amino acid composition and content of *Gonostegia hirta* (mg/g DW).

Compounds	Content
Aspartic acid (Asp)	0.288 ± 0.029
Serine (Ser)	0.062 ± 0.007
Glutamic acid (Glu)	0.043 ± 0.016
Glycine (Gly)	0.007 ± 0.002
Histidine (His)	0.091 ± 0.012
Arginine (Arg)	0.057 ± 0.023
Threonine (Thr)	0.033 ± 0.005
Alanine (Ala)	0.077 ± 0.023
Proline (Pro)	0.085 ± 0.006
Cysteine (Cys)	0.014 ± 0.009
Tyrosine (Tyr)	0.197 ± 0.013
Valine (Val)	0.020 ± 0.001
Methionine (Met)	0.015 ± 0.004
Lysine (Lys)	0.014 ± 0.001
Isoleucine (Ile)	0.009 ± 0.001
Leucine (Leu)	0.011 ± 0.002
Phenylalanine (Phe)	0.011 ± 0.000
Total	1.034

Note: “DW”: dry weight.

**Table 3 plants-12-00875-t003:** Fatty acid composition and content analysis of *Gonostegia hirta* (μg/g DW).

Compounds		Content
Octanoic acid	C8:0	0.7 ± 0.1
Decanoic acid	C10:0	0.8 ± 0.1
Lauric acid	C12:0	6.9 ± 0.7
Myristic acid	C14:0	59.0 ± 2.0
Pentadecanoic acid	C15:0	5.0 ± 0.2
Palmitoleic acid	C16:1	4.7 ± 0.6
Palmitic acid	C16:0	849.0 ± 30.0
Heptadecanoic acid	C17:0	9.7 ± 0.8
Linoleic acid	C18:2n6c	1898.0 ± 30.0
Linolenic acid	C18:3n6	1434.0 ± 36.0
Stearic acid	C18:0	215.0 ± 12.0
Stearolic acid	-	4.0 ± 0.9
Nonadecanoic acid	C19:0	22.0 ± 6.0
Arachidonic acid	C20:0	48.0 ± 6.0
Eicosanoic acid	C21:0	10.0 ± 2.0
Behenic acid	C22:0	65.0 ± 7.0
SFA ^a^ (%)		27.4
PUFA ^b^ (%)		72.6
Total fatty acids		4629.0

Note: ^a^ SFAs: saturated fatty acids; ^b^ PUFAs: polyunsaturated fatty acids; “DW”: dry weight.

**Table 4 plants-12-00875-t004:** The contents of TP, TF, and TS in *Gonostegia hirta* (mg/g DW).

Bioactive Compounds	Content
Total phenols	1.20 ± 0.1
Total flavonoids	76.49 ± 5.58
Total saponins	60.30 ± 0.66

Note: “DW”: dry weight.

**Table 5 plants-12-00875-t005:** The antioxidant activities of *Gonostegia hirta* extract.

Antioxidant Activity	Value
DPPH scavenging activity(mg/g)	5.92 ± 0.00
ABTS^+^ scavenging activity(mg/g)	68.82 ± 0.37
Ferric reducing power (μmol/L)	62.23 ± 0.80
Anti-superoxide anion viability (U/g)	11.82 ± 0.18
Inhibition of hydroxyl radicals (U/g)	90.90 ± 0.89

## Data Availability

The data presented in this study are available.

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
