# Peer review of "Nutritional Composition and Antioxidant Activity of Gonostegia hirta: An Underexploited, Potentially Edible, Wild Plant"

_plants, 2023, doi:10.3390/plants12040875_

Round 1
Reviewer 1 Report
Topic is interesting, and manuscript is well prepared and organized.
Results are presented and discussed in an appropriate manner.
Obtained results give a good picture and valuable information about nutritional properties of the studied plant.
English might be improved.
Introduction
The introduction could be amended with photos of the plant.
Expression 'heavy metals' should be replaced with 'potentially toxic and toxic elements'.
For better understanding I recommend that authors consider following paper:
“Heavy metals”—a meaningless term?
(IUPAC Technical Report), by John H. Duffus, Pure Appl. Chem., Vol. 74, No. 5, pp. 793–807, 2002
http://publications.iupac.org/pac/74/5/0793/index.html
M&M section
Line 333: More detailed description of the plants should be provided, i.e. time of harvesting.
Authors should describe what parts of the plant were used for analyses.
A photo of the plant would be very useful.
Line 337: Detailed procedure for digestion and ICP analysis should be provided.
Type of the instruments used and manufacturer should be provided
Results
Line 157: In the Figure 1 capture, what are N & D?
Line 186: trans-4-hydroxy-l-proline (Hyp), N-Acetylneuraminic acid (NANA), L-theanine, proline betaine
Correct it to: trans-4-hydroxy-l-proline (Hyp), N-Acetylneuraminic acid (NANA), L-theanine, proline betaine
Line 270: DW – should be explained
Line 305: FW should be explained
Line 315: VC and VE need full names first
Conclusions
It should outline the most significant results of the study.
Other comments included in the PDF file.

Author Response
Dear Reviewer, Thank you very much for your time involved in reviewing the manuscript and your very encouraging comments on the merits. We discuss each of your comments individually along with our corresponding responses

Reviewer 2 Report
The work sent to me for review is very interesting and well written. Its scientific value would also be highly appreciated if it were not for basic methodological shortcomings. The authors did not provide information on the stage of development of the plant. The chemical composition of plants changes significantly with the progress of growth, development, and especially flowering and maturation. In the discussion of the results, reference should be made to this issue. Authors should also provide information on the size of samples taken and the date of their collection (month and year). In addition, the work lacks such basic information as dry matter content, total protein content and fat content.
Other remarks
1. The authors should mention in the Introduction that Gonostegia hirta does not have an English name.
2. Since Gonostegia hirta is a little-known plant, it would be good to include a photo of it.
3. Table 1. Remove (mg/g DW) from the content of the table and enter it in the title as it is done in the other tables. Do not put a full stop after table titles.
4. I really like Figure 1, but it's too small and therefore illegible.
5. The title of table 5 looks strange.
6. Please remember to attach a supplementary file (Table 1S) to the publication.
7. Note that after citing the type of Hong et al. the publication number must be given in parentheses.
8. Correct puncutation errors throughout the text.
Author Response

(The authors gave the same response as above.)
